# Improved Quality Management of the Indian Meal Moth, *Plodia interpunctella* (Hübner) (Lepidoptera: Pyralidae) for Enhanced Efficacy of the Sterile Insect Technique

**DOI:** 10.3390/insects14040344

**Published:** 2023-03-31

**Authors:** Md. Mahbub Hasan, Md. Akhtar Hossain, Christos G. Athanassiou

**Affiliations:** 1Department of Zoology, Rajshahi University, Rajshahi 6205, Bangladesh; 2Laboratory of Entomology and Agricultural Zoology, Department of Agriculture, Crop Production and Rural Environment, University of Thessaly, Phytokou Str., 38446 Volos, Greece

**Keywords:** *Plodia interpunctella*, induced sterility, mating competitiveness, flight performance, DNA damage

## Abstract

**Simple Summary:**

The Indian meal moth, *Plodia interpunctella*, is an important pest of stored products across the globe. The sterile insect technique (SIT) has been widely used for controlling insect pests, but there are few reports on the use of the SIT against stored product insects. In this study, we tested parameters that can be used with success for the implementation of the SIT for the control of *P. interpuntella.* The current work indicated that the utilization of SIT for this purpose is feasible, and its efficacy is highly influenced by the irradiation dose, the life stage of the target species and the temperature level. Furthermore, the present findings also showed that the simultaneous release of sub-sterile male and sterile female moths might improve the efficacy of SIT, increasing the overall pest population suppression.

**Abstract:**

The sterile insect technique (SIT) is considered an environmentally friendly, autocidal control tactic to manage insect pests. This work dealt with the improvement of quality management of the Indian meal moth *Plodia interpunctella* (Hübner) for enhanced efficacy of the SIT. The results indicated that egg hatching of irradiated mature eggs of *P. interpunctella* was higher than that of younger eggs, indicating that mature eggs were significantly more tolerant than younger eggs. Moreover, our data revealed that a dose of 500 Gy completely prevented pupal formation in irradiated young and mature larvae. Crosses between irradiated and non-irradiated adults resulted in considerable variations in fecundity. The mating competitiveness index (CI) value was higher for a ratio of 5:1:1 (sterile male, fertile male, and fertile female, respectively) as compared with the ratio 1:1:1 for the irradiated individuals of all life stages. Low temperature (5 °C) maintenance of irradiated pupae significantly affected adult emergence. Using cylinders to assess flight ability, we found that the flight performance of adults that were developed from cold treated irradiated pupae was influenced by cylinder diameter, cylinder height and the number of hours the insects were in the cylinders. The percentage of DNA damage of the reproductive organs of adults developed from cold treated pupae that were irradiated with 100 and 150 Gy varied significantly. The results of this study should be used to implement pilot-scale field tests aiming at a sterile- to-fertile male ratio of 5 to 1.

## 1. Introduction

The Indian meal moth, *Plodia interpunctella* (Hübner) (Lepidoptera: Pyralidae) is a cosmopolitan insect pest infesting a wide range of food commodities which include nuts, pulses, meals, dried fruits and processed foods. It is considered by many researchers to be the most important moth pest of stored products globally [1]. The phasing out of the conventional fumigant methyl bromide following the Montreal Protocol will affect management programs for insect pests of stored products, including *P. interpunctella,* and will accelerate demand for new control strategies [2,3]. The management of stored product pests has been rapidly changing from an insecticide-based system to a more integrated approach [4,5]. Some insecticides, such as organophosphate and carbamate compounds, have been withdrawn from the post-harvest market, while others are under threat of removal. Moreover, certain strains of *P. interpunctella* have developed resistance to a wide range of insecticides, including biopesticides [5,6,7,8,9,10,11,12,13,14,15,16,17], although there is no direct evidence of field resistance to the latter [18]. These limitations necessitate an evaluation of alternative methods that can be successfully used for the control of this species without the need for insecticides.

One of these methods is the sterile insect technique (SIT), an environmentally friendly autocidal control tactic that has been used with great success in recent decades to manage insect pest populations on an area-wide basis [19]. The SIT requires the mass-rearing of the target insect, the sterilization of the male sex, and the sustained release of the sterile males over the target area at numbers large enough to overflood the wild insect population. A virgin wild female that has mated with a released sterile male will produce no offspring, and as a result, there will be fewer individuals in the population of the next generation [20]. The sequential release of competitive sterile males therefore reduces the reproductive potential of the wild female population, resulting in population suppression and in some instances, even eradication of the target population [21].

Lepidoptera are among the most radiation-resistant order of insects known, and applying a radiation dose to obtain full sterility in adult male moths would result in low quality insects [22]. Therefore, a derivative of the SIT called the inherited sterility technique (IS) or F1 sterility would be more efficient. In this technique, sub-sterile male moths exposed to lower gamma irradiation doses are released, and the mating of a sub-sterile male with wild female results in some offspring, but these are sterile [23].

One of the primary prerequisites for successful application of the SIT is the availability of a thriving mass-reared colony which is both reproductively compatible and competitive with the targeted wild population [24,25]. However, the process involved in the development of a strain for the SIT includes the initiation of a colony from wild-collected individuals, the laboratory colonization of the strain, the development of mass production methods, the development of an efficient gender separation (sexing) system that would allow the release of only sterile males, optimal sterilization/irradiation methods and the adoption of suitable handling, release and monitoring methods [26,27,28]. The insect laboratory colonization and mass-rearing processes can be considered some of the most important criteria for successful implementation of the SIT in the target areas; the successful use of the strain will largely depend on their mating performance and competitiveness after release into the field [29].

Although Cobalt-60 is the earliest and most widely used radioactive source in SIT, with advantages of good penetration performance and dose rate stability, the use of these sources in SIT projects as become more challenging recently, in view of stringent import regulations and high purchase costs. [30].

Despite the relatively wide range of data for the quality management of different insect pest species for the implementation of SIT-based protocols, there is still inadequate information in the case of *P. interpunctella*. Nevertheless, the use of the SIT in storage and processing facilities should be evaluated in more detail, given that the resources that are needed for such an application may be limited in comparison with insect pest management in field crops and orchards. Still, only a limited amount of data are available on the radiation sensitivity of certain life stages of *P. interpunctella*, as well as the subsequent mating behavior of the emerging sterile adult males. In view of the importance of this species in infestation of durable agricultural commodities throughout the world, and the limited knowledge on the effect of radiation for its control, the radiation sensitivity and performance of *P. interpunctella* were assessed. Our study had mostly focused on the effect of radiation on immature stages, *viz.* eggs, larvae and pupae, but we have also tested adult performance to a lesser extent..

## 2. Materials and Methods

### 2.1. Experimental Procedures

#### Insect Rearing

The *P. interpunctella* moths used in the study were derived from cultures maintained since 2014 at the Post-Harvest Entomology Laboratory of Rajshahi University, Rajshahi, Bangladesh. The colony was maintained on a standardized diet of corn meal, chick laying mash, chick starter mash (ingredients: ground corn, dehulled soybean meal, wheat middlings, rice bran, calcium carbonate, monocalcium phosphate, dicalcium phosphate, brewer’s yeast, vegetable oil, salt, choline chloride, ferrous sulfate, manganous oxide, zinc oxide, niacin supplement, vitamin E supplement, sodium selenite, copper sulfate, calcium pantothenate, ethylenediamine dihydriodide, biotin supplement, menadione sodium bisulfate complex, pyridoxine hydrochloride, riboflavin supplement, vitamin B12 supplement, folic acid, vitamin A supplement, vitamin D3 supplement, propionic acid (a preservative), diatomaceous earth, DL-methionine hydroxy analogue), and glycerol at a volumetric ratio (4:2:2:1), respectively [31]. The insect colony was maintained in an incubator at 27 °C and 70% relative humidity (RH), with a photoperiod of 16:8 (L:D) h.

### 2.2. Irradiation Procedure

Irradiation treatments were carried out using a Cobalt-60 source (Gamma cell-220, Nordian International Inc., Kanata, ON, Canada) located at the Atomic Energy Research Institute, Ganokbari, Savar, Dhaka, Bangladesh. Standard and routine dosimetry were carried out using the Fricke system (ISO/ASTM 51261). This dosimetry system was calibrated in accordance with the international standard ISO/ASTM 51261 [32]. The samples were placed at a distance of 50 cm from the gamma source and irradiated with a dose rate of 17.328 Gy/min, which was obtained from the source with activity of 38.62 kCi. The optimal radiation dose that induces sterility without compromising relative mating vigor was tested by submitting eggs, larvae and male pupae to different doses of gamma rays, and dose–response for mortality curves were established.

### 2.3. Irradiation throughout Ontogenesis

The doses selected for irradiation throughout ontogeny in *P. interpunctella* were based on a series of earlier studies [33,34]. Five replicate batches, each with 50 individuals of each stage, were used for each dose and the controls.

#### 2.3.1. Egg Irradiation

A total of 1280 either 1–2 d old (young) or 3–4 d old (mature) eggs were kept separately in 20 mL glass vials and irradiated in a calibrated ^60^Co gamma irradiator with the doses of 50, 75, 100 and 150 Gy. A group of untreated eggs served as a control group. The irradiated eggs were checked regularly and after hatching, the neonate larvae were transferred to transparent plastic rearing trays (9.6″L × 4.1″W × 2.0″H) (HL-B025, Jiangsu, China) where they were kept singly in 50 small holes (2 mL) that were filled with food medium (5 g) [31]. The insects were observed regularly until adult emergence. The percentage of adult emergence was recorded for each age and radiation dose.

#### 2.3.2. Larval Irradiation

A total of 1800 either neonate (1–2 d old) or mature (14–15 d old) larvae were irradiated with the doses of 50, 75, 100, 150, 300 or 500 Gy and kept separately in plastic rearing trays with food as described above. Three hundred untreated larvae were kept as a control group. The insects were observed regularly until adult emergence. The percentage of adult emergence was recorded for each larval age and radiation dose.

#### 2.3.3. Pupal Irradiation

A total of 2100 young (1–2 d) or mature (3–4 d) pupae were kept separately in plastic rearing trays as above, and irradiated with doses of 50, 75, 100, 150, 300, 400 and 500 Gy. A group of 300 untreated pupae served as a control group. After irradiation, they were observed daily for adult emergence, and the percentage of adult emergence for each age and dose was recorded. There were five replicates, each with 50 individuals of each stage, used for each dose and the controls.

### 2.4. Induced Sterility and Mating Competitiveness

#### 2.4.1. Crossing Schedule

To assess induced sterility and mating competitiveness of sterilized males developing from eggs and pupae and to assess the effect of increased sterilized male release ratios on egg hatching, competitiveness assays were completed.

#### 2.4.2. Irradiated Eggs

A total of 1000, 3–4 d old (mature) eggs were irradiated in the same irradiator with a dose of 150 Gy. Immediately after irradiation, the eggs were transferred individually in plastic rearing trays with food, as described above. The adults that developed from the irradiated eggs were mated in two combinations: treated male and treated female (TM by TF) and treated male and untreated female (TM by UF). Five replicates, each with fifty individuals, were used for each cross. The total number and viability (hatching) of the eggs oviposited by the female adults developing from the crosses with adult males emerging from the irradiated eggs were recorded for F1 generation.

#### 2.4.3. Irradiated Larvae

A total of 500 mature larvae were kept individually in cells of a plastic rearing tray with food and irradiated with 150 Gy when they were 3–4 days old. The adults that developed from irradiated larvae were mated and the egg viability was recorded for the F_1_ generation. There were two combinations: treated male and treated female (TM by TF) and treated male and untreated female (TM by UF).

#### 2.4.4. Irradiated Pupae

The sex of the pupae was determined based on the exogenital organs [35]. A total of 1500 pupae (1 and 2 d old) of each sex were kept separately in a plastic rearing tray. After irradiation with a dose of 150 Gy, the trays were kept in an incubator for adult emergence. The adults resulting from irradiated pupae were mated in two combinations: treated male and treated female (TM by TF) and treated male and untreated female (TM by UF). There were five replicates, each with ten pairs of male and female for each cross schedule and a control. The total number and viability (hatching) of the eggs per female adult developing from the irradiated pupae were recorded.

#### 2.4.5. Overflooding Ratio

Three different ratios were used for assessing the mating competitiveness of sterilized males: (irradiated male: unirradiated male: unirradiated female) 0:1:1, 1:1:1 and 5:1:1. The stages of *P. interpunctella* were selected for the overflooding ratio, as was carried out for the cross-schedule experiments. Two doses, 100 and 150 Gy, were also selected for irradiation. Mating schemes for all the ratios were carried out separately in plastic containers (500 mL) (RFL Plastic Container manufacturer, Rangpur, Bangladesh), and there were five replicates, each with ten pairs for each dose and ratio. The total number and viability (hatching) of the eggs laid by per female adults in the different treatments were recorded. The competitiveness index (C) was calculated following the method described by Fried [34].

### 2.5. Low-Temperature Treatment of Male Pupae and Mating Performance

Tests were carried out to assess the impact of cold storage of irradiated pupae on adult mating performance. Cold storage is often used to transport pupae from a rearing facility to a field release program. Mature larvae were put in plastic rearing trays with food, and after pupation, the young pupae (1–2 d) were irradiated with either 100 or 150 Gy. The irradiated pupae and control batch were maintained at 5 °C for 5- and 10-d, and after emergence, mating schedules (UM × UF, TM × UF, TM × TF) for all the overflooding ratios described above were carried out separately in plastic containers (250 mL) (RFL Plastic Container manufacturer, Rangpur, Bangladesh). There were five replicates, each with ten pairs for each dose and ratio. The total number and viability (hatching) of the eggs per female adults developing from the different stages were recorded.

### 2.6. Flight Cylinder Bioassay

In this trial as a quality test, the flight performance of adult males was assessed using different types of flight cylinders to detect quality difference between adult males that had emerged from irradiated (100 and 150 Gy) pupae that had been stored for 5 d at 5 °C, and non-irradiated pupae. The flight cylinders were 4, 6, 8 and 16 cm tall and were produced from PVC irrigation pipes (color = dark grey; diameter = 10 and 15 cm) [36]. The cylinders were placed on tables in a sealed room (30 × 35 ft^2^), and the interior surface of the cylinders was coated with talc to prevent the moths from crawling out of the cylinders. Male moths that had emerged from cold-treated and irradiated and unirradiated (control batch) pupae were placed inside a single flight cylinder. The number of moths remaining in each cylinder was recorded after a period of 0, 24, 48 and 88 h. There were five replications, each with ten adults for each dose, cylinder diameter and cylinder height. The bioassay was carried out at 27 ± 2 °C, 65 ± 5% RH and natural day and night light.

### 2.7. Assessment of DNA Damage in Cold Preserved Irradiated Pupae

DNA damage in individual cells of the reproductive organs of adult moths that had emerged from pupae maintained for 5 d at 5 °C and irradiated with 100 and 150 Gy was assessed using a DNA comet assay [37]. The reproductive organs of adults were dissected and frozen in liquid nitrogen, kept separate for each dose, and then gently homogenized into a powder, suspended in cold phosphate-buffered saline (PBS) and filtered through a 125-mm nylon mesh. A neutral comet assay was then used on the resulting cell suspension following the methods described by Hasan et al. [37]. The percentage of DNA damage was calculated with the following formula, as described by Sriussadaporn et al. [38]:$$\mathrm{DNS}\;\mathrm{damage}\;\%=\;\frac{\sum_{i=1}^{n}\left\{T\left(T+H\right)\right\}i}{n}\;\text{×}\;100$$
where ***T*** is the tail area of the comet image, ***H*** is the head area of the comet image, and ***n*** is the number of comet images.

### 2.8. Statistical Analysis

Levene’s [39] test was used to test the assumptions of normality and homogeneity of variance prior to statistical analysis and then the data were subjected to analysis of variance (ANOVA) using the PROC.ANOVA [40]. The data of percentage values were arcsine-transformed for statistical analysis. Means were separated by Tukey’s HSD test when the F-test of the ANOVA was significant at the 5% level. Untransformed means and standard errors are reported to simplify interpretation. The dose-response mortality for different stages was calculated using PROC.PROBIT [40]. For all experiments, the competitiveness index was calculated to determine the performance control and release sterile males using the Fried’s competitiveness index [34].

## 3. Results

### 3.1. Radiation Tolerance throughout Ontogeny

Egg hatching showed significant differences with dose in both young (F = 1128.86; df = 4,20; *p* < 0.0001) and mature eggs (F = 129.40; df = 4,20; *p* < 0.0001) (Figure 1). Mature eggs were significantly (F = 112.94; df = 5,44; *p* < 0.0001) more radio-tolerant than young eggs as indicated by higher egg hatching (Figure 1). A dose of 100 Gy and above completely prevented egg hatching when irradiated as young eggs (Figure 1).

Larval mortality varied significantly with dose in both young (23 d old) (F = 247.18; df = 6,28; *p* < 0.0001) and mature larvae (F = 187.06; df = 6,28; *p* < 0.0001) (Figure 2). A dose of 500 Gy completely prevented pupal formation in both larval age groups. More than 50% of the larvae developed into a pupa for doses of 50 and 75 Gy. Less than 50% of larvae developed into pupae with doses of 100 Gy or above, with the exception of mature larvae that were treated with 300 Gy. The percentage of pupal formation also varied significantly (F = 70.35; df = 7,62; *p* < 0.0001) between young and mature larvae. Adult emergence from pupae that were irradiated as young (F = 322.31; df = 6,28; *p* < 0.0001) or mature larvae (F = 468.73; df = 6,28; *p* < 0.0001) varied significantly with dose (Figure 3). A dose of 300 and 500 Gy administered to both young and mature larvae, prevented adult emergence. Irradiated mature larvae were more radiation tolerant than irradiated young larvae, in that they produced significantly (F = 327.25; df = 7,62; *p* < 0.0001) more pupae than the latter at all doses above 100 Gy (Figure 3). Pupae irradiation: adult emergence from irradiated pupae was also dose dependent (Figure 4) for both young (F = 302.71; df = 7,32; *p* < 0.0001) or mature pupae (F = 506.29; df = 7,32; *p* < 0.0001). In addition, late pupae (3–4 d old) were significantly (F = 246.47; df = 8,71; *p* < 0.0001) more tolerant to irradiation than young pupae (2–3 d old). A dose of 400 Gy prevented adult emergence from young pupae while 13.4% of adults emerged from mature pupae (Figure 4). 

The probit results indicated that young pupae were more tolerant to irradiation with respect to survival, as indicated by the higher values (360.53 Gy) for 99 percent mortality while the lowest (154.84 Gy) was recorded for young eggs (Table 1). The probit data also showed heterogeneity between all the stages irradiated as the Chi-square values are highly significant, which indicates a poor fit between the probit model and observed data (Table 1).

### 3.2. Induced Sterility and Mating Competitiveness

Significant differences were observed in the fecundity of the various crosses of adults that developed from irradiated late eggs (F = 1977.71; df = 2,12; *p* < 0.0001), mature larvae (F = 114.89; df 2,12; *p* < 0.0001) and young pupae (F = 154.45; df = 2,12; *p* < 0.0001) (Table 2). The significantly lowest fecundity was recorded in the irradiated males and irradiated female cross. The cross of irradiated males and irradiated females resulted in the lowest numbers of mature larvae compared to crosses in late eggs and pupae (Table 2). In addition, there were significant variations in the fecundity levels among all stages tested (F = 4.62; df = 2,43; *p* = 0.015). 

The induced sterility for all the crosses of adult males that developed from irradiated late eggs (F = 974.21; df = 2,12; *p* < 0.0001), mature larvae (F = 1733.63; df = 2,12; *p* < 0.0001) and pupae (F = 3691.11; df = 2,12; *p* < 0.0001) varied significantly (Table 2). In the treated male x untreated female cross, 100% sterility was recorded for the adults derived from irradiated pupae, while the respective figure for late eggs was 92% (Table 2). However, the induced sterility for the crosses did not vary significantly (F = 0.14; df = 2,42; *p* = 0.87) among stages. 

The mating competitiveness of the irradiated laboratory males was CI = 1 for the adults developed from mature larvae and pupae irradiated with 150 Gy. The CI was lower than 1 for males that developed from mature eggs and were therefore classed as less competitive (Table 2). 

The reproductive potential of the males that developed from irradiated (100 Gy) late eggs (F = 1345.87; df = 3,16; *p* < 0.0001), mature larvae (F = 714.63; df = 3,16; *p* < 0.0001) and pupae (F = 626.56; df = 3,16; *p* < 0.0001) (Table 3) varied significantly in overflooding ratio. The lowest fecundity was recorded in the case of the overflooding ratio of 5:1:1 (irradiated males: normal males; normal female) for all the life stages tested (Table 3).

For the 150 Gy treatment group, fecundity varied significantly with the overflooding ratio for adults that developed from irradiated mature eggs (F = 1658.17; df = 3,16; *p* < 0.0001), mature larvae (F = 847.61; df = 3,16; *p* < 0.0001) and pupae (F = 345.87; df = 3,16; *p* < 0.0001) (Table 3). Comparing however, the fecundity of the different stages, it was similar for the overflooding ratios at both doses, i.e., 100 Gy (F = 0.70; df= 2,57; *p* = 0.50) and 150 Gy (F = 0.28; df = 2,57; *p* = 0.75) (Table 3). 

The percentage sterility of adults varied significantly with increasing overflooding ratios of irradiated males that developed from mature eggs (F = 311.62; df = 3,16; *p* < 0.0001), mature larvae (F = 304.19; df = 3,16; *p* < 0.0001) and pupae (F = 218.75; df = 3,16; *p* < 0.0001) (Table 3). The same trend was observed in the 150 Gy treatment group for mature eggs (F = 455.09; df = 3,16; *p* < 0.0001), mature larvae (F = 435.10; df = 3,16; *p* < 0.0001) and pupae (F = 272.30; df = 3,16; *p* < 0.0001). However, comparing the percentage of sterility for all development stages, it did not vary significantly with overflooding ratios at both doses, i.e.,100 Gy (F = 0.29; df = 2,57; *p* = 0.49) and 150 Gy (F = 0.11; df = 2,57; *p* = 0.89). 

The mating competitiveness index (CI) was higher for all development stages when the overflooding ratio was 5:1:1 as compared with the ratio 1:1:1 for both irradiation doses (Table 3). The mating competitiveness index was equal to 1 for laboratory male adults developed from pupae and mature larvae irradiated with 100 Gy and 150 Gy with an overflooding ratio of 5:1:1. However, the males in the other groups were less competitive with CI values lower than 1 (Table 3). The lowest CI value (0.21) was recorded for adults that developed from pupae irradiated with 150 Gy in the 1:1:1 overflooding ratio.

### 3.3. Low-Temperature Treatment of Pupae and Male Mating Performance

The number of adults developing from 100 and 150 Gy-irradiated young pupae and that were maintained at 5 °C for 5 or 10 d was significantly reduced as compared with non-irradiated cold-preserved young pupae (F = 183.27; df = 3,56; *p* < 0.0001) (Figure 5). In addition, adult emergence of the 150 Gy treatment group was significantly lower than that of the 100 Gy treatment group (F = 230.98; df = 2,56; *p* < 0.0001). Adult emergence was also significantly affected by the duration of the cold treatment, i.e., it was lower in the 10 d treatment as compared with the 5 d treatment (F = 87.86; df = 1,56; *p* < 0.0001. In addition, the percentage of adult emergence was decreased at 150 Gy for both durations of preservation (Figure 5). There was also a significant (F = 39.64; df = 3,26; *p* < 0.0001) effect on the reproductive potential in the adults developing from pupae irradiated at 100 and 150 Gy and preserved at 5 °C for 5 d (Figure 6).

### 3.4. Flight Cylinder Bioassay

The cylinder diameter, cylinder height and the number of hours greatly influenced the flight performance of adult moths that had developed from irradiated (100 and 150 Gy) young pupae, maintained at 5 °C for 5 d (Figure 7 and Figure 8). The flight ability (propensity) of adult male moths developed from 100 Gy-irradiated pupae and maintained at 5 °C for 5 d was significantly better from flight cylinders that had a 10 cm diameter as compared with 15 cm (F = 16.20; df = 3,68; *p* < 0.0001) after 24 h (Figure 7). In addition, the mean number of male moths that failed to fly from the flight cylinders increased significantly with increasing height of the flight cylinder (F = 38.68; df = 3,68; *p* < 0.0001) (Figure 7). Additionally, the number of male moths developed from cold preserved irradiated pupae at 100 Gy that failed to fly from the cylinders significantly (F = 138.07; df = 3,152; *p* < 0.0001) decreased over time for all cylinder heights. Nevertheless, the diameter of the cylinder did not significantly (F = 0.01; df = 1,152; *p* = 0.90) influence the flight performance at 100 Gy. On the other hand, diameter (F = 47.69; df = 1,152; *p* < 0.0001), height (F = 83.49; df = 3,152; *p* < 0.0001) and time (F = 146.77; df = 3,152; *p* < 0.0001) significantly affected the flight performance of *P. interpunctella* adults that developed from cold-preserved pupae irradiated with 150 Gy (Figure 8).

### 3.5. Assessment of DNA Damage in Cold Treated and Irradiated Pupae

Figure 9 summarizes the percentage of DNA damage of the reproductive organs of adults from cold-treated pupae that were irradiated with 100 and 150 Gy. The percentage of DNA damage varied significantly (F = 108.81; df = 2,72; *p* < 0.0001) among the dose groups. The percentage of DNA damage increased as the radiation dose increased, suggesting that radiation tolerance is dose-dependent. In addition, both doses of gamma radiation caused a higher percentage of DNA damage after irradiation compared with control (Figure 10).

## 4. Discussion

Despite the fact that there are previous studies that have examined the performance of *P. interpunctella* following irradiation, the results of the present study illustrate specific interactions between the life stages that were irradiated and production of F1 progeny by the emerged adults. At the same time, we examined the effect of the combination of cold treatment and irradiation on the performance of the insects. Although the SIT has been thoroughly tested and implemented for several insect species that occur in the field, such as plant pests or public health pests [41], the stored-product facilities seem ideal for this technique, as there is no possibility of wild females invading the treated area from surrounding zones. As such, the vast majority of the irradiated males that will be released in a given storage or processing area will eventually remain at that area, competing with wild males. At the same time, considering the size of the facility, the calculation of the optimum number of irradiated males might be more accurate than in the case of other ecosystems. Finally, given that this is a more site-targeted application that does not require releases of large numbers of sterile individuals, the overall cost of such a post-harvest SIT protocol may be sufficiently reduced, as compared with other SIT-based control scenarios. All the above warrant additional experimentation on the basis of the feasibility of such an application.

Although irradiation was generally successful when applied at the egg stage, there were considerable variations between the two egg age groups used. In this context, mature eggs were more radiation-tolerant than younger eggs, a phenomenon that has also been recorded in the case of fumigants that are currently in use in stored-product protection [42,43].

It has been reported that for certain Lepidoptera pests, the effects of mutations that are caused by irradiation can be expressed during embryogenesis, but can also be expressed at a later stage [44,45]. On the other hand, despite differences in egg mortality, the immature development was heavily affected in both age groups tested, following a dose–response pattern. Hence, irradiation of eggs may be desirable for practical reasons, considering the increased cost of the application of irradiation at a later stage, and the fact that it can lead to considerable deformities and mortality levels in *P. interpunctella*.

Larvae were heavily affected by irradiation, even with the lowest dose; again age was important, given that young larvae were more susceptible and resulted in fewer pupae, following a dose–response pattern. The same holds in the case of pupae, as older pupae were less susceptible than younger ones. As the pupal stage is technically one of the solutions suggested to sex the irradiated individuals before their release, thereby minimizing the limitations of the presence of females, irradiation at that stage might be an important step towards designing mass production strategies [19]. It has been reported that both the optimal life stage and radiation dose are critical for the SIT [46]. For instance, the timing of pupal irradiation affects the quality of the resultant adults in oriental fruit flies *Bactrocera dorsalis* (Hendel) [47]. The findings of Jiang et al. also showed that the 8th day is the optimal age for irradiation of male pupae of codling moth *Cydia pomonella* (L.). Jiang et al. [48] reported that irradiation with 50–400 Gy had no significant effect on adult emergence in *Spodoptera frugiperda* (Smith), but that females were more sensitive than males in terms of reproductive parameters, especially when doses of radiation were >350 Gy. Nevertheless, in contrast with other pest groups such as Diptera, there is still inadequate information on the utilization of pupae in irradiation programs for other insect orders including Lepidoptera. This is a critical parameter that in the case of moths has largely relied on manual sex separation procedures [49]. To our knowledge, the only genetic sexing systems that have been developed so far for this purpose for moths are ones that have been reported for the silkworm, *Bombyx mori* (L.) (Lepidoptera: Bombycidae) [50] and the Mediterranean flour moth, *Ephestia kuehniella* Zeller (Lepidoptera: Pyralidae) [51], but not for *P. interpunctella*. In this context, this can be considered as a priority, especially when a regular check is needed to evaluate genetic breakdown [51]. However, the poor fit of the probit model, as indicated in Table 1, clearly suggests that there is a considerable percentage of variability that cannot be explained with the current relationship, indicating the complexity in the interaction of the factors that affect the performance tested here.

As expected, progeny production capacity was drastically reduced when irradiated males were mated with untreated females, which was further reduced in the case of mating of irradiated males with irradiated females. This was expressed much more vigorously as compared with mating of adults that had been irradiated at their immature stages. Still, irradiation may be technically complicated at the adult stage, compared with irradiation at the immature stages, in terms of mass production of irradiated individuals. On the other hand, we found that the closer the irradiation to the adult stage is, the higher the progeny production capacity. Similar results have been also reported for the tsetse fly, *Glossina austeni* Neustead (Diptera: Glossinidae) [52], in which irradiation, up to a certain dose, did not affect the mating performance of the males produced. In our study, despite the fact that we did not perform any direct comparisons in the mating performance between irradiated and non-irradiated males, we also found that the males produced after irradiation were less competitive at elevated irradiation doses. Still, the “critical balance” between producing *P. interpunctella* males that are 100% sterilized and males that are 100% competitive compared with normal males was not found in the doses tested here; however, our results suggest that such a balance could be optimized between 100 and 150 Gy. Indirectly though, we were able to carry out a quality check based on our bioassay protocols [52]. In part, these dissimilarities can be alleviated by the changes in the ratio indicated here. However, sterility was not affected by the *P. interpunctella* life stage, for both doses examined, in the case of irradiated males. Moreover, the flight cylinder bioassay has shown that, for some of the treatments tested, the compatibility was considerable, especially at 100 Gy. The results of our histological studies confirm additional evidence that the mating performance of the irradiated adults was not much affected, at least based on the morphological characteristics used as indicators here. Despite this, the DNA data have shown that there was considerable damage, although there were no morphological differences in the reproductive organs of the exposed individuals. Nevertheless, our primary goal in this work was to investigate the effect of the treatments tested on immature development of the target species, and not much on adult performance, and hence very little can be concluded regarding the competitiveness of irradiated adults. Moreover, the flight performance tests that were carried out in our work were mostly focused on flight initiation, and not the efficacy of flight. Additional experiments are needed, using a wind tunnel, to explore this aspect.

Exposure or even “storage” at low temperatures has been thoroughly tested to preserve insects with desired characteristics, which can be then utilized in pest management programs. For instance, Haque et al. [53] used cold storage of eggs of *P. interpunctella* for mass rearing protocols for the egg parasitoid *Trichogramma evenescens* Westwood (Hymenoptera: Trichogrammatidae), with good survival rates and sufficient preservation times. Based on the data reported here, sterility was considerable when the adults emerged from irradiated pupae that were exposed at 5 °C and kept for 5 d. Consequently, these conditions can be further evaluated in mass-rearing protocols, in terms of prolonging the interval for which the irradiated individuals can be resealed, with no loss in efficacy. Nevertheless, there were some negative interactions in the performance of the adults developing from different irradiated stages and cold storage, which could be attributed to the effect of cold on *P. interpunctella* pupae, which are considered sensitive to low temperatures [54]. In an earlier study, Athanassiou et al. [55], found that pupae of this species can be almost totally controlled after exposure for 3 d at −5 °C.

## 5. Conclusions

In summary, our study provides the inferences necessary for the development of an SIT-based control strategy in *P. interpunctella*. In contrast with similar species for which SIT-based strategies have been demonstrated, the management of this species has some incontestable advantages from a practical point of view, as (i) it does not require the large insect numbers that are needed in the case of pests in the open field, as such an application can occur in within a storage or a processing facility; (ii) it may require a lower number of augmentative releases of sterile males; (iii) it does not involve the negative influence of the invasion of mated and fertile females from the surrounding environment; and (iv) it may be easier in terms of release of the desired individuals, as this does not require aerial release techniques (airplanes, etc.), but can be performed from ground stations. The poor fit of the probit model underlined the fact that predicting adult performance from irradiated immature subjects may not be always feasible, on the basis of deviations from standard mating behavior. All of the above can be further examined with additional experimental work in semi-field or even field studies.

## Figures and Tables

**Figure 1 insects-14-00344-f001:**
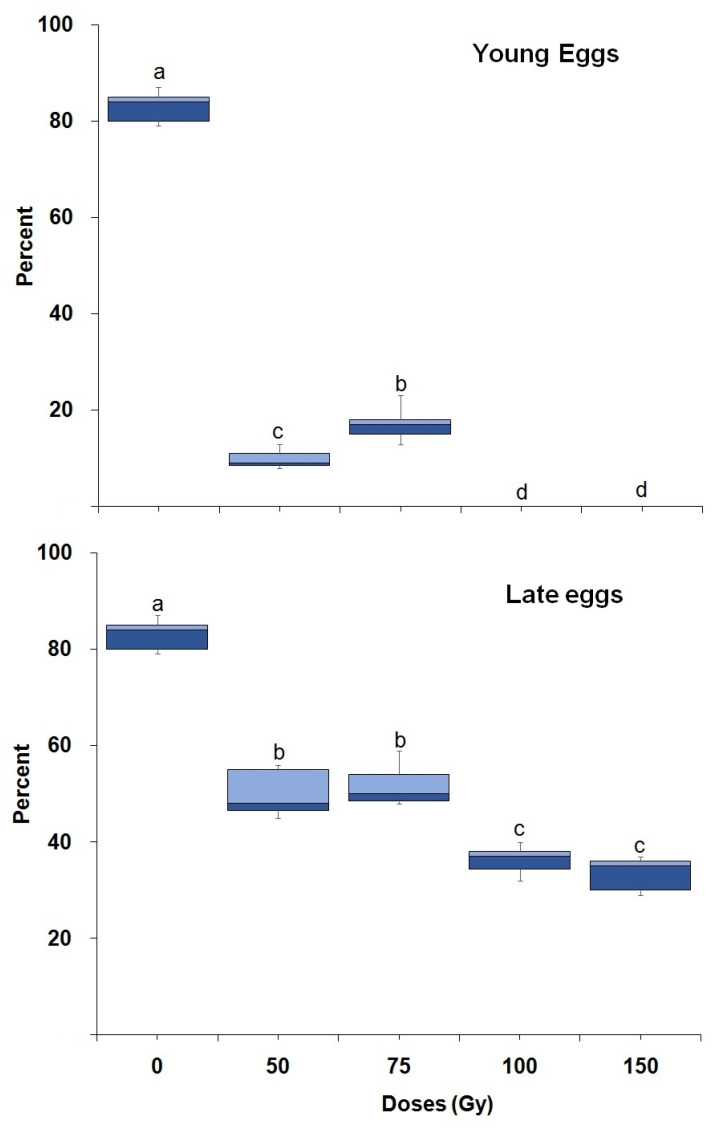
Percentage hatching of *P. interpunctella* eggs irradiated as young and late eggs. Boxes at each dose and age followed by the same letter do not differ significantly; HSD test at 0.05.

**Figure 2 insects-14-00344-f002:**
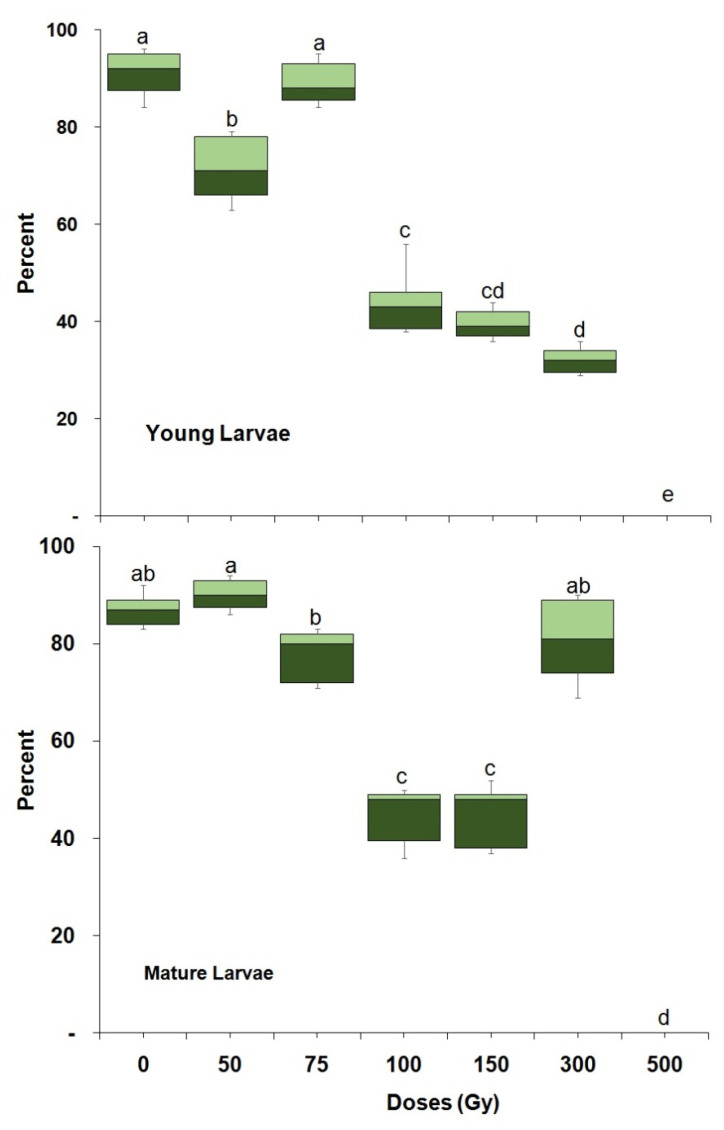
Percentage of *P. interpunctella* pupae developing from larvae irradiated at a young (2–3 d old) and late age (mature). Boxes at each dose and stage followed by the same letter do not differ significantly; HSD test at 0.05.

**Figure 3 insects-14-00344-f003:**
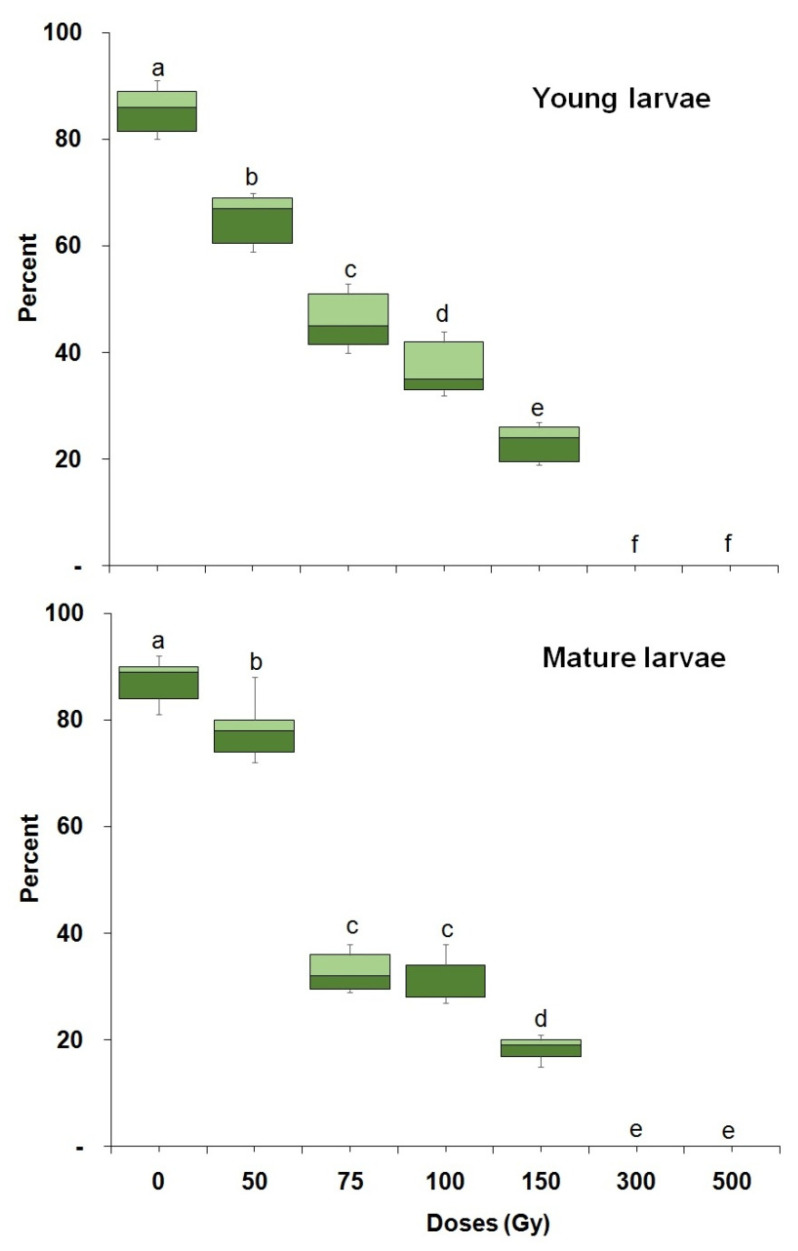
Percentage of *P. interpunctella* adult emergence from pupae from larvae irradiated at a young and late age. Boxes at each dose and stage followed by the same letter do not differ significantly; HSD test at 0.05.

**Figure 4 insects-14-00344-f004:**
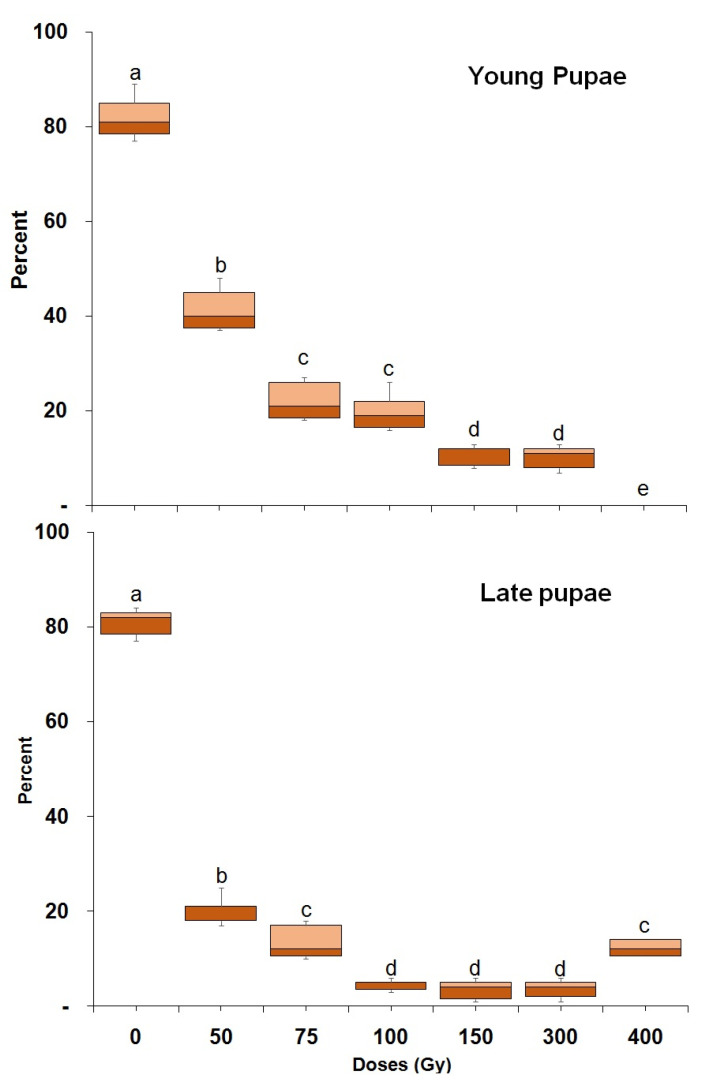
Percentage of *P. interpunctella* adult emergence from pupae irradiated at a young and late age. Boxes at each dose and stage followed by the same letter do not differ significantly; HSD test at 0.05.

**Figure 5 insects-14-00344-f005:**
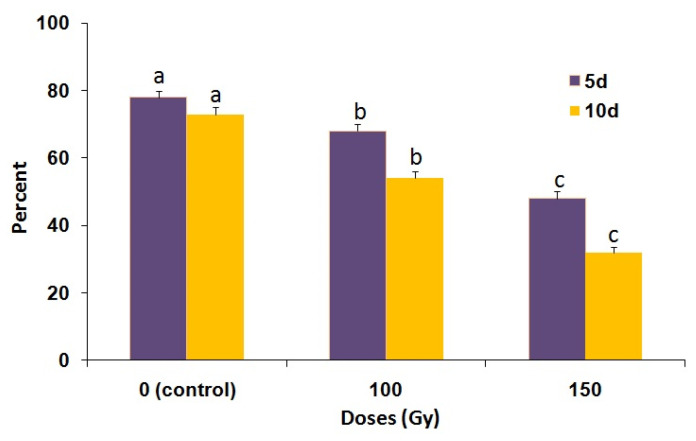
Percentage of adult emergence of *P. interpunctella* developing from irradiated pupae preserved at 5 °C for 5 and 10 days. Bars within each dose and day followed by the same letter do not differ significantly; HSD test at 0.05.

**Figure 6 insects-14-00344-f006:**
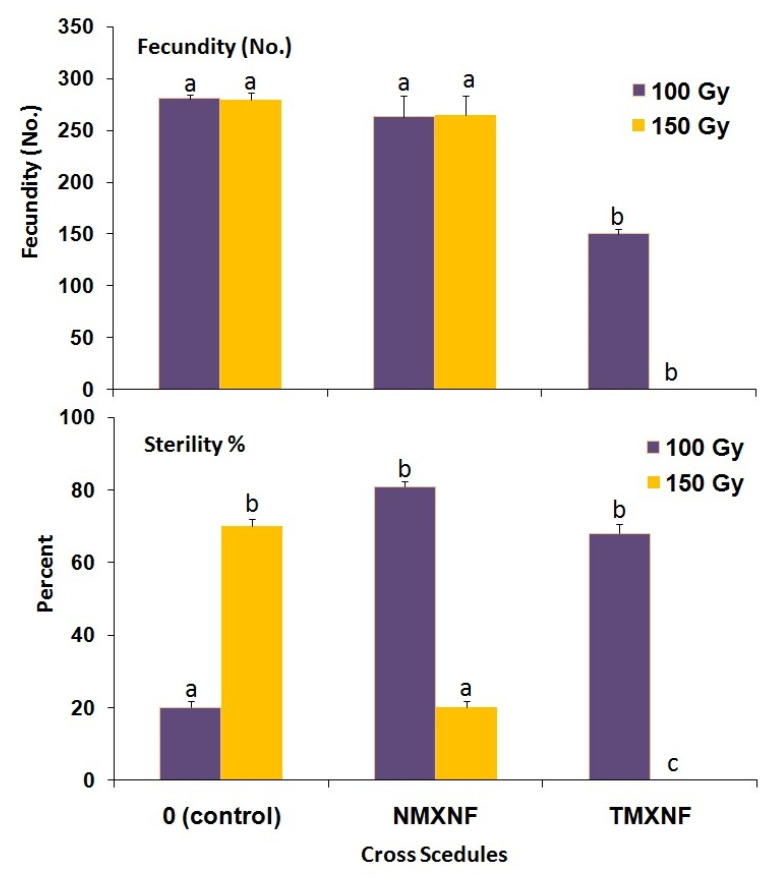
Fecundity and percent sterility for different cross schedules of *P. interpunctella* developing from irradiated pupae preserved at 5 °C for 5 days. Bars for each dose and cross followed by the same letter do not differ significantly; HSD test at 0.05.

**Figure 7 insects-14-00344-f007:**
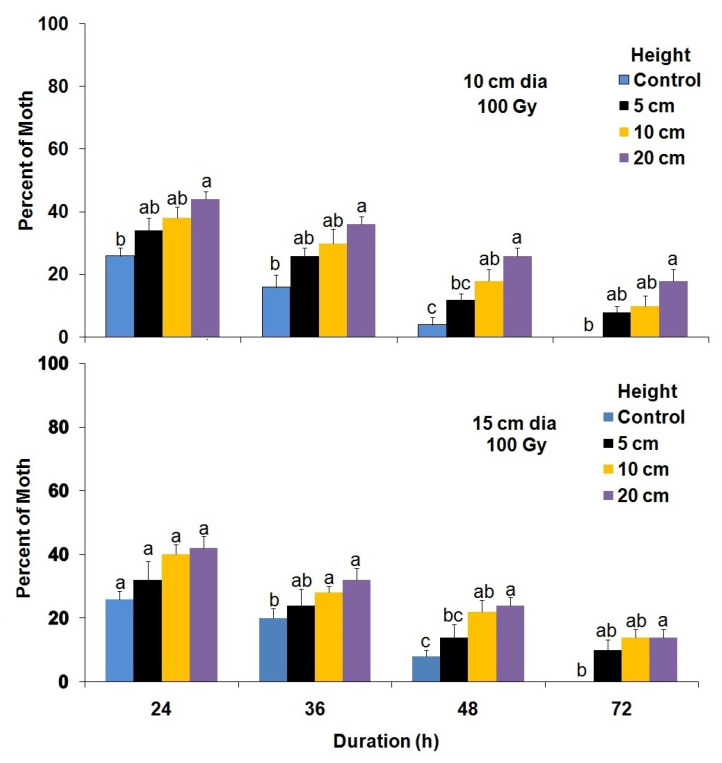
The mean number (±SE) of *P. interpunctella* male moths developing from cold-preserved (5 °C) pupae irradiated at 100 Gy that failed to fly from the flight cylinders at different interval periods for each height. Bars for each duration followed by the same letter do not differ significantly; HSD test at 0.05.

**Figure 8 insects-14-00344-f008:**
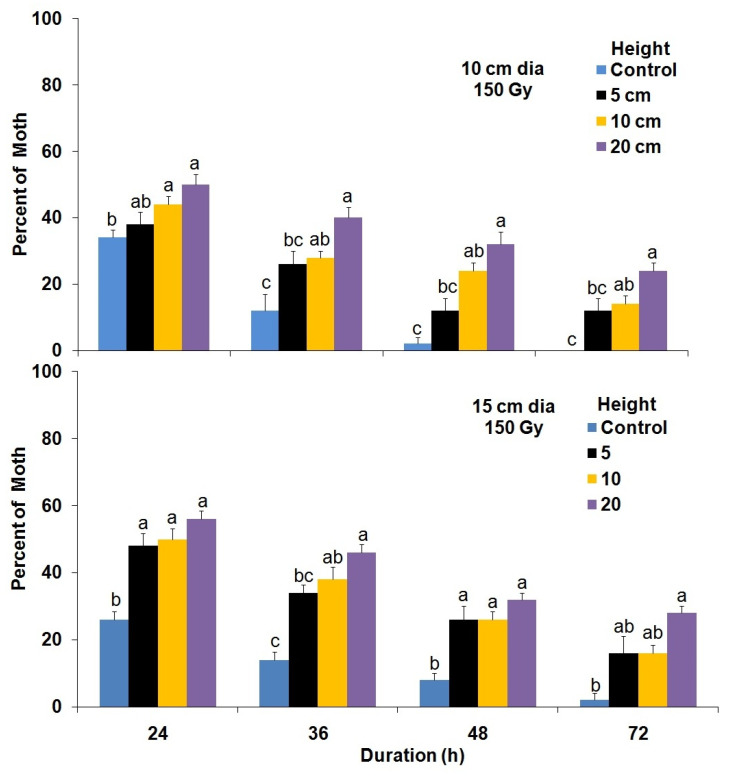
The mean number (±SE) of *P. interpunctella* male moths developing from cold-preserved (5 °C) pupae irradiated at 150 Gy that failed to fly from the flight cylinders at different interval periods for each height. Bars for each duration followed by the same letter do not differ significantly; HSD test at 0.05.

**Figure 9 insects-14-00344-f009:**
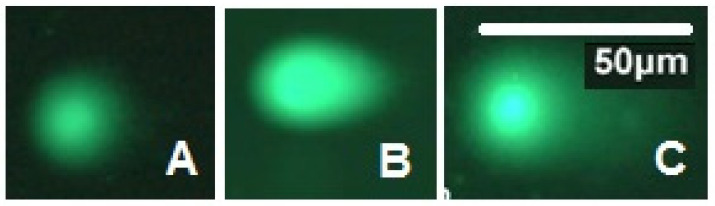
Typical DNA comet from adult *P. interpunctella* reproductive organs developing from irradiated pupae preserved at 5 °C for 5 d. ((**A**)—Control; (**B**)—100 Gy; (**C**)—150 Gy).

**Figure 10 insects-14-00344-f010:**
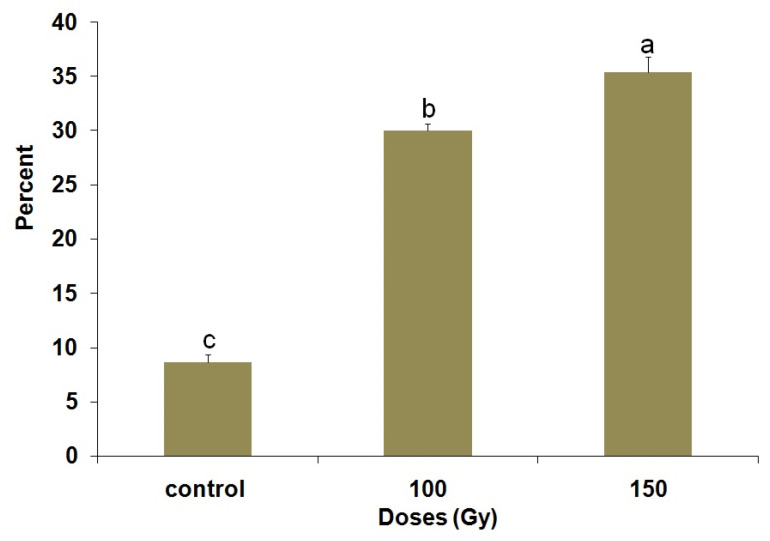
Gamma radiation-induced DNA damage percent in adult *P. interpunctella* reproductive organs developing from irradiated pupae preserved at 5 °C for 5 d. Bars followed by the same letter do not differ significantly; HSD test at 0.05.

**Table 1 insects-14-00344-t001:** Probit analyses of mortality for different ages of *P. interpunctella* eggs, larvae and pupae treated with gamma radiation.

Stages	N	LC_50_ (Gy) (95% Fiducial Limits)	LC_99_ (Gy) (95% Fiducial Limits)	Slope ± SE	Intercept ± SE	χ^2^ Values
Young eggs	1280	21.37 (13.67–39.22)	154.84 (98.90–216.34)	0.66 ± 0.12	−27.73 ± 20.77	151.88 (*p* < 0.001) (df,10)
Late eggs	1280	59.36 (30.42–76.11)	206.18 (159.27–315.10)	1.23 ± 0.16	−44.67 ± 17.58	36.26 (*p* < 0.001) (df,10)
Young larvae	1800	51.48 (37.75–61.78)	299.29 (227.85–486.96)	0.03 ± 0.56	165.33 ± 55.83	55.01 (*p* < 0.001) (df,16)
Mature larvae	1800	58.89 (49.59–66.29)	230.87 (193.38–302.78)	0.06 ± 0.55	163.71 ± 54.89	42.62 (*p* < 0.001) (df,16)
Young pupae	2100	26.20 (19.34–41.08)	360.53 (211.24–437.14)	0.09 ± 0.37	98.14 ± 31.21	72.29 (*p* < 0.001) (df,16)
Late pupae	2100	16.19 (11.27–25.31)	343.92 (296.16–448.02)	0.06 ± 0.55	−63.18 ± 26.31	39.18 (*p* < 0.001) (df,16)

**Table 2 insects-14-00344-t002:** Induced sterility and mating competitiveness index of different *P. interpunctella* cross schedules irradiated at 150 Gy.

Cross Schedules	Late Eggs			Mature Larvae			Pupae		
	Fecundity (No.)	Sterility%	CI*	Fecundity (No.)	Sterility%	CI	Fecundity (No.)	Sterility%	CI
UF × UM	186.20 ± 2.62 ^a^	18.40 ± 1.2 ^b^		173.73 ± 3.73 ^a^	18.80 ± 1.5 ^c^		284.80 ± 5.99 ^a^	19.40 ± 1.33 ^b^	
TM × UF	158.00 ± 4.71 ^c^	92.40 ± 1.4 ^a^	0.75	125.00 ± 3.02 ^b^	95.40 ± 1.0 ^b^	1.00	166.20 ± 7.44 ^c^	100 ± 00 ^a^	1.01
TM × TF	137.80 ± 2.36 ^b^	87.60 ± 1.3 ^a^		99.20 ± 3.82 ^c^	100 ± 00 ^a^		146.20 ± 8.17 ^b^	100 ± 00 ^a^	

UM—untreated male; TF—treated female; UF—untreated female; TM—treated male; CI*—competitiveness index. Means within a column followed by the same letter do not differ significantly; HSD test at 0.05.

**Table 3 insects-14-00344-t003:** Fecundity, induced sterility and mating competitiveness index of sterile *P. interpunctella* males in laboratory conditions measured with different ratios of sterile to untreated males (I_m_:U_m_:U_f_).

Doses (Gy)	Flooding Ratio I_m_:U_m_:U_f_	Late Eggs			Mature Larvae			Pupae		
		Fecundity (No.)	Sterility%	CI*	Fecundity (No.)	Sterility%	CI	Fecundity (No.)	Sterility%	CI
	0:1:1	256.80 ± 4.08 ^a^	20.00 ± 0.84 ^c^	-	240.00 ± 4.64 ^a^	16.80 ± 1.89 ^c^	-	272.60 ± 5.62 ^a^	20.20 ± 0.86 ^d^	-
	1:0:1	121.30 ± 2.39 ^b^	51.60 ± 1.97 ^b^	-	139.20 ± 5.66 ^b^	47.40 ± 2.39 ^b^	-	193.80 ± 4.04 ^b^	48.40 ± 3.57 ^c^	-
100	1:1:1	114.60 ± 3.24 ^c^	88.40 ± 2.61 ^a^	0.37	121.80 ± 3.94 ^c^	92.00 ± 2.94 ^a^	0.28	55.00 ± 4.86 ^b^	83.60 ± 2.93 ^a^	0.42
	5:1:1	107.90 ± 4.24 ^d^	93.40 ± 1.97 ^a^	0.93	102.40 ± 4.54 ^c^	96.20 ± 1.07 ^a^	0.70	50.20 ± 2.23 ^c^	98.20 ± 0.80 ^a^	1.00
	0:1:1	256.40 ± 1.73 ^a^	20.60 ± 0.68 ^d^	-	268.40 ± 4.88 ^a^	18.20 ± 0.86 ^c^	-	280.40 ± 6.21 ^a^	17.00 ± 0.71 ^c^	-
	1:0:1	109.60 ± 3.55 ^b^	63.80 ± 1.28 ^c^	-	127.80 ± 3.39 ^b^	43.20 ± 3.32 ^b^	-	179.60 ± 3.85 ^b^	58.20 ± 4.74 ^b^	-
150	1:1:1	98.20 ± 4.96 ^c^	87.60 ± 2.95 ^b^	0.29	115.60 ± 2.98 ^c^	94.20 ± 1.69 ^a^	0.43	149.00 ± 3.78 ^c^	99.20 ± 0.38 ^a^	0.21
	5:1:1	87.80 ± 3.60 ^c^	100 ± 00 ^d^	0.72	113.80 ± 3.64 ^d^	100 ± 00 ^a^	1.0	127.80 ± 5.43 ^d^	100 ± 00 ^a^	0.53

CI*—competitiveness index. Means within a column for each dose followed by the same letter do not differ significantly; HSD test at 0.05. I_m_—irradiated male; U_m_—unirradiated male; U_f_—unirradiated female.

## Data Availability

All data are contained within the article.

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
