# Peer review of "Improved Quality Management of the Indian Meal Moth, Plodia interpunctella (Hübner) (Lepidoptera: Pyralidae) for Enhanced Efficacy of the Sterile Insect Technique"

_insects, 2023, doi:10.3390/insects14040344_

Round 1

Reviewer 1 Report (Previous Reviewer 2)

The minimal revisions have been made, and I agree the paper can be published.  Some text editing may still be necessary.

Author Response

Reviewers 1

Comments and Suggestions for Authors

The minimal revisions have been made, and I agree the paper can be published.  Some text editing may still be necessary.

Response: Thanks for the comments. We have revised the ms according to reviewer comments.

Reviewer 2 Report (New Reviewer)

Title: Add a comma and a space between moth and Plodia.

Simple Summary:

Use past tenses in the following sentences:  “the The current work indicates that the utilization of SIT for this purpose is feasible, and there are specific biological traits that can determine the success of such an application in storage and processing commodities, such as the irradiation dose, the life stage of the species that is to be treated, and low temperature maintenance of the insects.”

Abstract:

Use past tenses in the following sentences:  “This work deals with the improvement of quality management of the Indian meal moth Plodia interpunctella (Hübner)for enhanced efficacy of the SIT.The results indicate that egg hatch of irradiated mature eggs of ...”

Introduction:

Use past tenses or past perfect tenses in the following sentence:  “Our study is mostly focused on the effect ...”

Materials and Methods:

There are many words stuck to each other without spaces, for example 1280either (p. 4, line 125), please separate the words.

 Results:

There are many words stuck to each other without spaces, for example 100Gy (p. 6, line 238), please separate the words.

Discussion:

Please highlight and mention the novelty of this study.

 Conclusions:

Move these sentences to the Discussion section:  However, the poor fit of the probit model, as indicated in Table 1, clearly suggests that there is a considerable percentage of variability that cannot be explainedwith the current relationship, indicating the complexity in the interaction of the factors that affect the performance tested here.

Author Response

Reviewers 2

Comments and Suggestions for Authors

Title: Add a comma and a space between moth and Plodia.

Response: Has been corrected as suggested.

Simple Summary:

Use past tenses in the following sentences:  “the The current work indicates that the utilization of SIT for this purpose is feasible, and there are specific biological traits that can determine the success of such an application in storage and processing commodities, such as the irradiation dose, the life stage of the species that is to be treated, and low temperature maintenance of the insects.”

Response: Has been corrected as suggested.

Abstract:

Use past tenses in the following sentences:  “This work deals with the improvement of quality management of the Indian meal moth Plodia interpunctella (Hübner)for enhanced efficacy of the SIT.The results indicate that egg hatch of irradiated mature eggs of ...”

Response: Has been corrected as suggested.

Introduction:

Use past tenses or past perfect tenses in the following sentence:  “Our study is mostly focused on the effect ...”

Response: Has been corrected as suggested.

Materials and Methods:

There are many words stuck to each other without spaces, for example 1280either (p. 4, line 125), please separate the words.

Response: Has been corrected as suggested.

 Results:

There are many words stuck to each other without spaces, for example 100Gy (p. 6, line 238), please separate the words.

Response: Has been corrected as suggested.

Discussion:

Please highlight and mention the novelty of this study.

Response: Has been corrected as suggested.

 Conclusions:

Move these sentences to the Discussion section:  However, the poor fit of the probit model, as indicated in Table 1, clearly suggests that there is a considerable percentage of variability that cannot be explained with the current relationship, indicating the complexity in the interaction of the factors that affect the performance tested here.

Response: Has been corrected as suggested.

Reviewer 3 Report (New Reviewer)

I always appreciate the studies focusing on non-chemical strategies against insect pest control. Therefore, I personally admire this study. However, it is always important to understand whether the proposed control method is relevant to crop production, storage, and market access. For example, Australia adopts a strategy like nil tolerance for live insects in exported food and grains, thus it is less likely to be usefulness of applying the SIT strategy against stored insect pests in the Australian context. But it might be worthwhile study for some commodities in different parts of world. In overall, the manuscript is roadworthy, but significant modification is required to make it publishable.

Simple summary:

L11: Should have a space between genus and species, like Plodia interpunctella, not joining each other.

L12: SIT strategy is for the control, not the management.

In overall, the summary is too vague, rather it should be much precise and clear.

Abstract:

L22: Concise sentence is always preferable, for example on an area-wide basis (too long) which should be like extensively.

L29: Should write like 5:1:1 (sterile male, fertile male, and fertile female respectively).

Introduction:

L41: cosmopolitan insect pest (Should be added insect in the middle).

L80: should be as the.

L85: as should be removed.

L98: Immature? Objectives are not clear yet.

Materials and methods:

L105: Chicken starter mash? What does it contain? Please provide more detail.

L106: Better put the ratio inside parentheses.

L121: Please write this method in detail, referring to the previous papers is not enough.

L125: Please correct typo- 1280either (put space between).

L129: where were- remove the repetitive word. So many typos, so please proofread well the manuscript.

L131: Until adult emergence? Write time period exactly, be exact quantitative, not the qualitative.

L156: Five replicates each having fifty- sound awkward sentence. Please rewrite.

L223: Statistical analysis: provided very generic explanation. Please write in detail that which datasets undergone through which analysis or correction or whatever. Be specific.

Results: Be consistent with colour of graphs.

L499: Conclusion- conclusion is supposed to be purely based on your study or main finding from your study.

Final comments: This study significantly consists of basic experiments undertaken. The study is very interesting, although the study is solely laboratory based. The spectrum of research experiments carried out in this study and novelty deserve the paper to be publishable. But, unless the significant improvement on grammar, structure, and typos, the manuscript is not believed to be read worthy. Still there is lack of harmony among research objectives, materials and methods, and result interpretation.  

Author Response

Reviewers 3

Comments and Suggestions for Authors

Simple summary:

L11: Should have a space between genus and species, like Plodia interpunctella, not joining each other.

Response: Correction has been made.

L12: SIT strategy is for the control, not the management.

Response: Has been corrected as suggested.

In overall, the summary is too vague, rather it should be much precise and clear.

Response: This part was corrected, as suggested.

Abstract:

L22: Concise sentence is always preferable, for example on an area-wide basis (too long) which should be like extensively.

Response: This part was corrected as suggested. See changes throughout both summary and abstract.

L29: Should write like 5:1:1 (sterile male, fertile male, and fertile female respectively).

Response: Has been corrected as suggested.

Introduction:

L41: cosmopolitan insect pest (Should be added insect in the middle).

Response: Has been corrected as suggested.

L80: should be as the.

Response: Has been corrected as suggested.

L85: as should be removed.

Response: Has been corrected as suggested.

L98: Immature? Objectives are not clear yet.

Response: Has been revised as suggested.

Materials and methods:

L105: Chicken starter mash? What does it contain? Please provide more detail.

Response: Has been added as suggested.

L106: Better put the ratio inside parentheses.        

Response: Has been corrected as suggested.

L121: Please write this method in detail, referring to the previous papers is not enough.

Response: Has been added as suggested.

L125: Please correct typo- 1280either (put space between).

Response: Has been corrected as suggested.

L129: where were- remove the repetitive word. So many typos, so please proofread well the manuscript.

Response: Has been corrected as suggested.

L131: Until adult emergence? Write time period exactly, be exact quantitative, not the qualitative.

L156: Five replicates each having fifty- sound awkward sentence. Please rewrite.

Response: Has been corrected as suggested.

L223: Statistical analysis: provided very generic explanation. Please write in detail that which datasets undergone through which analysis or correction or whatever. Be specific.

Response: Has been corrected as suggested.

Results: Be consistent with colour of graphs.

Response: Has been corrected as suggested.

L499: Conclusion- conclusion is supposed to be purely based on your study or main finding from your study.

Response: Has been corrected as suggested.

This manuscript is a resubmission of an earlier submission. The following is a list of the peer review reports and author responses from that submission.

Round 1

Reviewer 1 Report

The paper “Improved quality management of the Indian meal moth Plodia interpunctella (Hübner) (Lepidoptera: Pyralidae) for enhanced efficacy of the sterile insect technique” reports the assessment of the radiation sensitivity and performance of Plodia interpunctella to Cobalt-60 source. The results are useful to provide a sustainable and effective strategy of area-wide management of P. interpunctella using SIT. Generally speaking, this manuscript is meaningful and deserves publication. However, some issues need to be explained or revised before this manuscript can be accepted. 

Why the authors used Cobalt 60 as irradiation source? Although Cobalt 60 is the earliest and most widely used radioactive source in SIT with advantages in good penetration performance and dose rate stability, a serious problem has arisen recently for the use of it in SIT projects as it is becoming almost impossible to acquire radioactive sources for insect sterilization as well as its radioactive pose a safety hazard. Moreover, the SIT program using Cobalt 60 as the radioactive source is also limited by economic constraints (Zhang et al., 2022. Bulletin of Entomological Research, DOI:10.1017/S0007485322000323). The authors should mention this information in the Introduction.

Optimal life stage for radiation and optimal dosage are critical for SIT. The timing of pupal irradiation affects the quality of the resultant adults (Fezza et al. 2021, Appl. Entomol. Zool. 56: 443-450). Recent studies have shown that young pupal stage is more sensitive to radiation, and has been observed in many species of lepidoptera, such as Cydia pomonella (Zhang et al., 2022. Bulletin of Entomological Research, DOI:10.1017/S0007485322000323) and Spodoptera frugiperda (Jiang et al., 2022, Pest Management Science, 78: 2806-2815). In this study, effects of radiation on each stages of P. interpunctella were assessed. I have no idea why the authors designed such an experiment. 

The paper must have information regarding mapping dose (dose distribution), size of irradiation chamber, area of maximum and minimum dose in the irradiation chamber or the container, throughput per hour or per day, other specific characteristics of the X ray generator (Voltage, current, and so on). 

The authors determined the effect of radiation on the flight performance of adult males using different types of flight cylinders, a simple device. I'm not sure how well of the P. interpunctella adults. It is suggested that the author adopt a flight mill to measure the detailed flight parameters, including flight times, flight distance, and flight velocity. 

Line 133-139. The pupal period of P. interpunctella is 7-14 days (Naeemullah et al., 1999. Applied entomology and zoology, 34(2): 267-276). So it is not accurate for mean 3-4 d pupae as mature pupae. 

Line 127-128. Lack a space between 1800 and either. 

Line 204-205. Lack a space between at and 27.

Author Response

Response to Reviewers comment

Reviewer 1

Comments: Why the authors used Cobalt 60 as irradiation source? Although Cobalt 60 is the earliest and most widely used radioactive source in SIT with advantages in good penetration performance and dose rate stability, a serious problem has arisen recently for the use of it in SIT projects as it is becoming almost impossible to acquire radioactive sources for insect sterilization as well as its radioactive pose a safety hazard. Moreover, the SIT program using Cobalt 60 as the radioactive source is also limited by economic constraints (Zhang et al., 2022. Bulletin of Entomological Research, DOI:10.1017/S0007485322000323). The authors should mention this information in the Introduction.

Response: We have agreed with the reviewer comments. However, we have used here Cobalt 60 as irradiation source since it is the only facility available in the institute. We have added this information in the introduction.

Comments: Optimal life stage for radiation and optimal dosage are critical for SIT. The timing of pupal irradiation affects the quality of the resultant adults (Fezza et al. 2021, Appl. Entomol. Zool. 56: 443-450). Recent studies have shown that young pupal stage is more sensitive to radiation, and has been observed in many species of lepidoptera, such as Cydia pomonella (Zhang et al., 2022. Bulletin of Entomological Research, DOI:10.1017/S0007485322000323) and Spodoptera frugiperda (Jiang et al., 2022, Pest Management Science, 78: 2806-2815). In this study, effects of radiation on each stage of P. interpunctella were assessed. I have no idea why the authors designed such an experiment. 

Response: Thanks for the comments. We have determined the optimal radiation doses throughout the ontogeny of P. interpunctella prior to select the doses for SIT, given that there are no any specific doses published for SIT in P. interpunctella. Moreover, the doses vary from species to species.

Comments: The paper must have information regarding mapping dose (dose distribution), size of irradiation chamber, area of maximum and minimum dose in the irradiation chamber or the container, throughput per hour or per day, other specific characteristics of the X ray generator (Voltage, current, and so on). 

Response: The additional information regarding the gamma source has been added in the text under head “2.2. Irradiation procedure”

Comments: The authors determined the effect of radiation on the flight performance of adult males using different types of flight cylinders, a simple device. I'm not sure how well of the P. interpunctella adults. It is suggested that the author adopt a flight mill to measure the detailed flight parameters, including flight times, flight distance, and flight velocity. 

Response: We determined only the flight ability of the adult P. interpunctella developing from irradiated pupae.

Comments: Line 133-139. The pupal period of P. interpunctella is 7-14 days (Naeemullah et al., 1999. Applied entomology and zoology, 34(2): 267-276). So it is not accurate for mean 3-4 d pupae as mature pupae. 

Response:We have checked the article in where they recorded as 7-9 days. We had gone through the article of Naeemullah et al., 1999, but there are certain variations since in that work the authors used rice bran as a diet. Moreover, the specific work is mostly focused on diapause, which is sufficiently differentthan the parameters used here.

Title: Relationship of cold tolerance to developmental determination in the Indian meal moth, Plodia interpunctella (Lepidoptera: Phycitidae), Applied Entomology and Zoology 34(2):267-276. DOI: 10.1303/aez.34.267

Comments: Line 127-128. Lack a space between 1800 and either. 

Response: Has been corrected as suggested.

Comments: Line 204-205. Lack a space between at and 27.

 Response:Has been corrected as suggested.

Reviewer 2 Report

This paper presents potentially valuable work into the use of SIT for Plodia interpunctella control, but suffers from a variety of flaws, including conceptual design as well as in analysis and presentation of data.  Sadly, there are many instances in the text where the presentation is quite careless. Some examples (but not a comprehensive list) are given below.

line 67.  a claim is made that Lepidoptera are notoriously "radiation resistant" but not a single reference is given.  How can such a strong assertion be made with no evidence to back it up?  

l. 78.  a statement of need is made for a sexing system for producing only sterile males.  Why, and what is this statement based on?  The biology of the insect is hardly mentioned or considered in this study, yet is key to determining the appropriate use of sterility.  What is known regarding the mating behavior of P. interpunctella, and why would releasing only sterile males be the effective and justified tactic? What could be expected from  releasing sterile females?   

This lack of background on mating behavior is also evident in the design of the mating tests given later on in the manuscript.  As the authors themselves admit, they do not carry out a direct comparison of the performance of irradiated vs. non-irradiated males (see Discussion, line 447).  Why not?? Successful SIT relies on thorough knowledge of mating strategies.  For example, in screwworm, the fact that females usually mate only once was a key finding which justified the mass release of sterile males.  A single mating between a fertile female and a sterile male would render all eggs from that female sterile.  What is the situation with Plodia and what is the biological basis for the authors' choice of experimental design and tests?  Again, no review of mating behavior is given.

l. 112.   "dose-response curves were established" but authors do not state what the response variables were.... The reader is forced to hunt around later in the manuscript to determine which variables were studied in relation to dose.

l. 169. how were these mating containers selected? why were sterile and fertile males tested together, yet never compared directly against each other? Why and how were the particular ratios chosen for mating tests? To make matters worse, explanation of the ratios is not always complete.  For example, in line 172, "5:1:1" is followed by "(sterile untreated male: untreated female)".  What about the third element in the ratio?

l. 192.  what does this flight bioassay really measure? how has it been used in other studies? No description of the behavior or its interpretation is given. 

lines 210-213. careless typing and proofing of the text is abundant, with lack of spaces between words occurring frequently ("frozenin liquid nitrogen", "kept separatefor each dose", etc.)

Examples of problems with careless, incomplete presentation of data and data analysis:

l. 236 Figure 1.  Missing information, as the "box and whisker" plots have no explanation; 2nd panel in the figure, with data for late eggs, has no letters for significance.  

Table 1.  All chi square values are highly significant, which indicate a poor fit between the probit model and observed data.  Yet, authors do not provide any discussion or interpretation at all. This is unacceptable. It's true that significant chi square can be a common occurrence and does not necessarily invalidate the probit analysis, but it should be addressed.  

l. 468. The genus of parasitic wasp is incorrectly written as Trochogramma, another example of careless writing.

In the end, I am left with many unanswered questions about the significance of this study.  Many of them arise from the lack of background on Plodia's natural biology and its mating behavior.  This study is not publishable unless it undergoes serious revision. 

Author Response

Response to Reviewers comment

Reviewer 2

Comments: line 67.A claim is made that Lepidoptera are notoriously "radiation resistant" but not a single reference is given. How can such a strong assertion be made with no evidence to back it up?  

Response: Reference has been added as suggested.

Comments: l. 78.  A statement of need is made for a sexing system for producing only sterile males.  Why, and what is this statement based on?  The biology of the insect is hardly mentioned or considered in this study, yet is key to determining the appropriate use of sterility.  What is known regarding the mating behavior of P. interpunctella, and why would releasing only sterile males be the effective and justified tactic? What could be expected from releasing sterile females?   

Response: It has been reported that the sterilized male insectsuse to mate more than once with their female counterparts subsequently the populations decline in a predictable manner. This is the reason to select the male for sterilization rather than female. The same approach has been also utilized in other pests as well, as it is shown in past reports.

Comments: This lack of background on mating behavior is also evident in the design of the mating tests given later on in the manuscript. As the authors themselves admit, they do not carry out a direct comparison of the performance of irradiated vs. non-irradiated males (see Discussion, line 447). Why not?? Successful SIT relies on thorough knowledge of mating strategies. For example, in screwworm, the fact that females usually mate only once was a key finding which justified the mass release of sterile males. A single mating between a fertile female and a sterile male would render all eggs from that female sterile. What is the situation with Plodia and what is the biological basis for the authors' choice of experimental design and tests? Again, no review of mating behavior is given.

Response: Our work was not entirely focused to the idea of studying the effect of SIT in the mating behavior of the irradiated individuals as a standalone parameter, but the interaction of certain additional factors, such as the dose, the sex ratio, the temperature etc. In this context, we have illustrated the effect of these factors in terms of survival, while “mating success” was an additional parameter that was tested in parallel.

Comments: l. 112.   "dose-response curves were established" but authors do not state what the response variables were.... The reader is forced to hunt around later in the manuscript to determine which variables were studied in relation to dose.

 Response: Has been corrected as suggested.

Comments: l. 169. how were these mating containers selected? why were sterile and fertile males tested together, yet never compared directly against each other? Why and how were the particular ratios chosen for mating tests? To make matters worse, explanation of the ratios is not always complete.  For example, in line 172, "5:1:1" is followed by "(sterile untreated male: untreated female)".  What about the third element in the ratio?

 Response: Has been corrected as suggested in 2.4.5. Overflooding ratio.

 Comments: l. 192.  What does this flight bioassay really measure? how has it been used in other studies? No description of the behavior or its interpretation is given.

 Response: The flight bioassay mainly concerns the flight activity as well as the ability of the sterilized individuals for movement (whichcan be concomitantly considered as an indicator of distribution to the target area).

 Comments: lines 210-213. careless typing and proofing of the text is abundant, with lack of spaces between words occurring frequently ("frozen in liquid nitrogen", "kept separatefor each dose", etc.)

Response: Has been corrected as suggested.

Comments: l. 236 Figure 1.  Missing information, as the "box and whisker" plots have no explanation; 2nd panel in the figure, with data for late eggs, has no letters for significance.  

 Response: Figure 1 has been corrected as suggested.

Comments: Table 1.  All chi square values are highly significant, which indicate a poor fit between the probit model and observed data.  Yet, authors do not provide any discussion or interpretation at all. This is unacceptable. It's true that significant chi square can be a common occurrence and does not necessarily invalidate the probit analysis, but it should be addressed.  

Response: The information has been mentioned in the text as suggested.

Comments: l. 468. The genus of parasitic wasp is incorrectly written as Trichogramma, another example of careless writing.

 Response: Has been corrected s suggested.

Round 2

Reviewer 1 Report

Although the authors have addressed most all of my concerns, some of the key issues were not substantially raised, and the suggested references were not cited and well discussed in the Introduction and Discussion.

Although Cobalt 60 is the earliest and most widely used radioactive source in SIT with advantages in good penetration performance and dose rate stability, a serious problem has arisen recently for the use of it in SIT projects as it is becoming almost impossible to acquire radioactive sources for insect sterilization as well as its radioactive pose a safety hazard. Moreover, the SIT program using Cobalt 60 as the radioactive source is also limited by economic constraints (Zhang et al., 2023. Bulletin of Entomological Research,  11372-78). The authors should mention this information in the Introduction.

Optimal life stage for radiation and optimal dosage are critical for SIT. The timing of pupal irradiation affects the quality of the resultant adults (Fezza et al. 2021, Appl. Entomol. Zool. 56: 443-450). Although the authors have determined the optimal life stage for radiation, this information should be mentioned and discussed, and some related literatures should be cited, for example, Zhang et al., 2023. Bulletin of Entomological Research,  11372-78; Jiang et al., 2022, Pest Management Science, 78: 2806-2815.

Author Response

Response to Reviewers comment (R2)

Reviewer 1

Comments: Although the authors have addressed most all of my concerns, some of the key issues were not substantially raised, and the suggested references were not cited and well discussed in the Introduction and Discussion.

Response: Thanks for your comments. All the queries have been now addressed as you suggested. 

Comments: Although Cobalt 60 is the earliest and most widely used radioactive source in SIT with advantages in good penetration performance and dose rate stability, a serious problem has arisen recently for the use of it in SIT projects as it is becoming almost impossible to acquire radioactive sources for insect sterilization as well as its radioactive pose a safety hazard. Moreover, the SIT program using Cobalt 60 as the radioactive source is also limited by economic constraints (Zhang et al., 2023. Bulletin of Entomological Research,  113: 72-78). The authors should mention this information in the Introduction.

Response: This information has been added in the introduction as you suggested. 

Comments: Optimal life stage for radiation and optimal dosage are critical for SIT. The timing of pupal irradiation affects the quality of the resultant adults (Fezza et al. 2021, Appl. Entomol. Zool. 56: 443-450). Although the authors have determined the optimal life stage for radiation, this information should be mentioned and discussed, and some related literatures should be cited, for example, Zhang et al., 2023. Bulletin of Entomological Research,  113: 72-78; Jiang et al., 2022, Pest Management Science, 78: 2806-2815.

Response: This information as well as references has been added in the discussion as you suggested. 

Reviewer 2 Report

The authors have made very minor corrections to the manuscript.  Aside from this,  they have limited themselves to repeating my comments in the text but have not addressed them.  They must provide a better explanation and justification of why mating behavior was not directly addressed or taken into account in the flight assays, and why sterile males were not directly compared to irradiated males.

Author Response

Response to Reviewers comment (R2)

Reviewer 2

Comments: The authors have made very minor corrections to the manuscript.  Aside from this, they have limited themselves to repeating my comments in the text but have not addressed them.  They must provide a better explanation and justification of why mating behavior was not directly addressed or taken into account in the flight assays, and why sterile males were not directly compared to irradiated males.

Response: Indeed, the experimental design could be based on the comparison of the behavior of the different crosses with the “control” adults, both in terms of negative and positive controls. Nevertheless, due to the fact that the current work was largely focused on the treatment of immature life stages of the target species, and the concomitant males, this is why our treatments were NM X NF vs TM X NF vs TM vs TF (and not NM vs TF). In the vast majority of studies with irradiated short-lived adults for different species, this is considered as a normal practice, given that irradiated males are the ones that are used in SIT-based programs (i.e. papers with fruitflies, mosquitoes etc.). We would like to underline that the comment by the reviewer is valuable and greatly contributes to the effort. This comment is greatly appreciated, and we will definitely take it into account in the next experimental designs.

Round 3

Reviewer 1 Report

The authors have addressed all my comments, and this manuscript should be accepted for publication in Insects.

Author Response

Response to Reviewers comment (R3)

Reviewer 1

Comments: The authors have addressed all my comments, and this manuscript should be accepted for publication in Insects.

Response: Thanks for the comments.

Reviewer 2 Report

Authors have acknowledged the reviewers' criticisms (not just mine, but the others as well) of the experimental design and analysis, yet they still do not fully reflect this in the text.  They should clarify explicitly in the text the limitations of this study:

* the focus is on the effects of radiation on immature stages, and very little can be gleaned from this study regarding the competitiveness of irradiated adults

* flight tests are only a test of whether the moths could leave the container, not whether their flight is effective. As another reviewer pointed out, a wind tunnel assay should be used for flight tests.

* given the poor fit of the probit model, which authors have acknowledged, what validity does the mortality study have?  Authors need to insert a statement on the use and validity of Table 1.

These additions to the text are fairly simple but necessary before the paper is published.

Author Response

Response to Reviewer comment (R3)

Reviewer 2

Authors have acknowledged the reviewers' criticisms (not just mine, but the others as well) of the experimental design and analysis, yet they still do not fully reflect this in the text.  They should clarify explicitly in the text the limitations of this study:

Comment: the focus is on the effects of radiation on immature stages, and very little can be gleaned from this study regarding the competitiveness of irradiated adults

Response:  I am agreed with the comments. However, the handling of the irradiated adult moths is difficult compared to immature stages. In addition, it has been reported that the pupal stage is more convenient to irradiate prior to releasing for SIT. In this point of view, we have mostly focused on the irradiation of immature stages rather than the adult stage. 

However, we have explained as well as added the following text regarding this information in the discussion section:

Nevertheless, our primary goal in this work was to investigate the effect of the treatments tested on the immature development of the target species, and not much on adult performance, and hence very little can be concluded regarding the competitiveness of irradiated adults.

Comment: Flight tests are only a test of whether the moths could leave the container, not whether their flight is effective. As another reviewer pointed out, a wind tunnel assay should be used for flight tests.

Response: we have performed experiments relating to the flight ability of the normal and irradiated moth. Moreover, we have followed the procedures as described by J. E. Carpenter, T. Blomefield & M. J. B. Vreysen 2012. A flight cylinder bioassay as a simple, effective quality control test for Cydia pomonella. Appl. Entomol. 136 (2012) 711–720. I am also agreed with the reviewer's comment since there are so many published articles relating to wind tunnel flight assay for semio-chemical and assessed the fitness of sterile males.

However, we have explained as well as added the following text regarding this information in the discussion section:

information in the discussion section:

Moreover, the flight performance tests that were carried out in our work were mostly focused on flight initiation, and not the efficacy of flight in terms of flight efficacy. Additional experimentation is needed towards this direction, with the utilization of a wind tunnel.

Comment: given the poor fit of the probit model, which authors have acknowledged, what validity does the mortality study have?  The authors need to insert a statement on the use and validity of Table 1.

Response: We have explained as well as added the following text regarding this information in the conclusion section:

However, the poor fit of the probit model clearly suggests that there is a considerable percentage of variability that cannot be explained with the current relationship, indicating the complexity in the interaction of the factors that affect the performance tested here